# Evaluation of the coordinated development between Chinese cultural industry and scientific & technological innovation

**Zhenni Yu**[1], **Jian Yu**[2,3]*

1 Chinese Academy of International Trade and Economic Cooperation, Beijing, 100710, PRC, 2 School of Economics and Management, Beijing Forestry University, Beijing, 100083, PRC, 3 Institute of Internet Industry, Tsinghua University, Beijing, 100085, PRC

* yujianbjfu@bjfu.edu.cn

**Data Availability Statement:** All relevant data are within the manuscript and its Supporting Information file.

**Funding:** This paper is supported from Soft Science Research Project of Beijing Science and

## Abstract

Based on the coupling and interaction relationship between China's cultural industry (CI) and scientific & technological innovation (STI), this study constructed an index system for their coordinated development. The weight of each indicator was determined by using the entropy value method (EVM), and the coupling coordination degree (CCD) model was used to calculate CCD and coordination degree of China's CI and STI from 2012 to 2020. On this basis, the key factors in the coupling effect were analyzed using grey correlation degree (GCD). The results demonstrate that: (1) there is a high-level coupling relationship between China's CI and technological innovation; (2) the level of coupling coordination between the two is generally on the rise, experiencing a development process from serious maladjustment to high coordination; (3) Industry resources, policy support and the cost of cultural undertakings are the endogenous factors restricting the development of CI, and the environment and output of STI are the key factors restricting the coupling and coordinated development of Chinese CI and STI.

## 1. Introduction

Under the background of economic globalization and the acceleration of industrial transformation in the world, cultural industry (CI) has gradually become a new pillar industry in many countries. In China, the development of CI has also risen to the level of national strategy: the development of CI during the 14th Five-year Plan period will be the beginning stage of realizing the strategic goal of becoming a cultural power by 2035, and the opening stage of comprehensively building a modern CI system and boosting the in-depth integration of "culture +".

All of these highlight the importance and necessity of developing CI, and the combination of "culture + science & technology (S&T)" has become a significant link to promote the development of China's CI. Under the general trend of industrial integration, the integration of CI and STI has mainly demonstrated its vigorous vitality. During the COVID-19 pandemic, China's digital economy played the role as an economic "stabilizer", promoting the equalization

Technology Plan "Research on Innovative Development Model of Science and Technology promoting Cultural Creative Industry" (#Z09031001590911). The funders had specify role in study design, data collection and analysis.

**Competing interests:** The authors have declared that no competing interests exist.

and diversification of residents' cultural consumption. With the advent of the post-epidemic digital economy era, culture and S&T are increasingly demonstrating a trend of integration, and their integration is gradually developing in depth [1]. Currently, the development of CI in China is principally due to the growth of modern power, the improvement of S&T, as well as the market space released by the reform of the cultural system.

There is a "dualistic dialectic" relationship between scientific & technological innovation (STI) and the development of CI. They influence and promote each other: under the background of STI, the development of CI not only requires the support of talents, but also needs to integrate the existing regional cultural resources to create an environment for the development of the emerging cultural formats. The interaction of S&T, culture and economy is the driving force for the development of modern civilization and CI in China. The integration of CI and S&T has formed a virtuous circle, and the important development trend of common development has injected new vitality into CI.2.

## 2. Literature review

The diversified demand, digital applications and immersive experiences are also driving the vigorous development of new business forms that integrate culture and technology [2]. Social and cultural progress is largely related to scientific and technological progress and the application of technologies in scientific development. S&T is an important driving force for the disruptive innovation effect of CI. Luo et al. [3] held that the construction of "Digital Culture in China" relies on the integration of culture and technology.

The support of S&T for cultural development is reflected in the sequential progression of material, method and concept [4]. At the same time, in the contemporary society, STI remains to be a utility and source of creativity for the creativity of cultural heritage [5], which not only enhances the production process, creative level and technical content but also enriches the display forms of culture, thus speeding up the dissemination of culture. At present, the integration of culture with S&T has achieved a crossover disruptive innovation with the help of Internet technology [6].

Many scholars have made useful explorations on the interaction and integrated development of CI and technological innovation. In the relevant academic circle, foreign scholars have conducted plenty of studies. Negroponte (1978) [7], the founder of MIT Media Lab, first proposed the phenomenon and concept of industrial convergence, which started the coupling study on culture and S&T. Reinganum [8] combined innovation with industry and proposed the theory of industrial innovation; meanwhile, from the perspective of industry, he took technological innovation as an important thrust for the optimization of industrial structure. Since then, scholars have gradually paid attention to this research field and started to conduct researches at the theoretical level [9] at the macro level. Sandelowski [10] pointed out that technology is the manifestation of culture, and STI is deeply embedded and rooted in the core of culture. Day et al. [11] found that culture and technology are mutually promoted. Berman et al. [12] further pointed out that science and technology can promote the innovation and growth of CI. Kehoe & Adebanjo [13] discussed the impact of STI on the quality of cultural output, and proved that STI is conducive to improving the quality of cultural output, gaining wider market, and thus promoting cultural innovation. Leidner & Kayworth [14] studied the interaction between information technology and culture and clarified their relationship—unity of opposites. Ibata-Arens & Lincoln [15] and other scholars discussed the positive role of STI in increasing the added value of cultural products, improving the core competitiveness of CI and promoting the development of CI; they held that STI can give birth to new cultural products and formats, change the forms of traditional cultural products, and optimize the

communication channels of cultural products. In addition, STI can improve the technicality and novelty of traditional cultural and creative products, completely changing the traditional cultural and creative industry, with an effect of being 0 to 1 [16]. Caiani [17] proposed that STI can help improve the speed and efficiency of industrial development, rationalization and sophistication of industrial structure, as well as labor productivity, thus fundamentally improving the quality of industrial development. At the micro level, some scholars began to conduct studies on a certain countries and regions: Saha [18], Koizumi [19]and other scholars examined the impact of Japanese traditional culture on STI and believed that culture plays a crucial role in the development of the STI in Japan.

In China, although CI started late, with the government's attention and rapid development of the industry in recent years, the integration of CI and STI has attracted more and more attention, and scholars have conducted a lot of research in this field. At the macro level, many Chinese scholars have summarized the modes and types of integration between culture and science and technology from different perspectives [1]. Hao et al. [20] based on the diamond model, proposed that the technological innovation capability of CI is the key driving force for the high-quality development and industrial competitiveness. In addition, some Chinese scholars came to the conclusion that technological innovation can provide technical support for CI [21] and upgrade the production efficiency [22]. But at the same time, some scholars held that CI and technological innovation are mutually reinforced and promoted [23], which has been recognized by more and more scholars in recent years. For example, Ding [24] pointed out that the coupling coordination between CI and STI can contribute to expanding the added value of CI, promoting STI, as well as promoting the transformation and upgrading of economic structure. In addition, some scholars measured and evaluated the degree of integration between culture and S&T by constructing the development index or index system of integration of culture and S&T [25] and explored the innovative development path of CI. At the micro level, Guo and Pan [26] analyzed the experience of the United States, the European Union and Japan in promoting "the coupling of culture and technology". Fan and Yin [27] studied the development path of CI in ethnic minority areas of Hubei Province from the perspective of STI, all of which provided useful reference for the transformation and upgrading of China's CI.

Combining the research at home and abroad, it can be found that the present research on the coupling of CI and STI are mainly using a qualitative approach, and the relevant empirical research is minor. Furthermore, the empirical research conducted from both the macroscopic and microscopic angles is scarce. In view of this, the present study constructed the evaluation systems of CI and STI, and reviewed the coupling coordination between two systems. The relationship between CI and STI was revealed from the macro level. Based on this, further analysis of the impact of specific STI indicators on CI was conducted to disclose the microscopic mechanism behind this coupling phenomenon from the index level. This study will be beneficial to the accurate evaluation of the development situation of CI and STI, providing reference for relevant departments in China taking specific measures to promote the coordinated development of the coupling system of culture and S&T, and further promote the coordinated development of China's CI and STI. This will boost the progress of China's CI and truly achieve cultural confidence, thus building a powerful cultural country.

## 3. Research hypothesis

In the existing research of the academic circle, scholars at home and abroad all believe that CI and STI are mutually coordinated and mutually promoted, and some factors in STI can promote the development of CI on a certain level. Currently, academic research views on the key

factors in the coupling effect of CI and STI can be broadly divided into four aspects: the scientific and technological innovation environment (STIE), the input of scientific and technological innovation (ISTI), the output of scientific and technological innovation (OTIE), and the performance of scientific and technological innovation (STIB).

CI and STI influence and promote each other. In the same economic system, the relationship between the two sub-industrial systems is not independent, such as CI and STI. On the one hand, with the increasing intensity and speed of the application of CI to STI, the cycle of STI is shortened accordingly. On the other hand, the rapid development of STI has realized the transition of technological track, thus promoting the evolution of CI to the direction of information, digitalization and high intensification. Specifically speaking, as a creative industry, the CI needs the achievements of STI as the material carrier to promote its development. Correspondingly, the CI can promote the progress of STI by providing spiritual motivation and other ways. Therefore, CI and STI have a deeply coupling effect. However, with the development of CI and STI becoming increasingly mature and our country vigorously promoting the integration of "culture + technology" industry gradually obtained results, the two also tend to be coordinated.

Based on the above analysis, a hypothesis is proposed:

*H1*: *The development of CI coupled with STI tends to be coordinated.*

The optimization of the STIE is closely related to the development of the CI. For CI and STI, the environment plays a certain intermediary role. The development of the CI and the development of STI are in the same environment, so the situation of the STIE can be seen from the side to reflect the environment of the CI. Moreover, culture is rooted in people's minds and can affect their initiative and tendencies. When the industrial theory of "culture + S&T" is rooted in people's ideas, people will be influenced by culture subtly and create and optimize the STIE. In other words, culture creates a positive ideological atmosphere and environment for the development of science and technology. Therefore, the STIE is itself a form of expression of the CI.

Based on the above analysis, another hypothesis is proposed:

*H2*: *The optimization of the STIE is closely linked to the development of the CI.*

There is a significant correlation between increased ISTI and the development of the CI. On the one hand, with the increasing penetration and integration of S&T in the development process of CI, the trend of digital culture, information technology and "Internet +" is becoming more and more obvious. The S&T elements of the CI have gradually increased, and the two have gradually converged. Therefore, a considerable part of the input of actual STI directly or indirectly plays a role in the development of CI. On the other hand, culture can influence the formulation of S&T policies and the development of STIE [28], so the development of CI can react to the ISTI.

Based on the above analysis, the third hypothesis is proposed:

*H3*: *There is a significant correlation between increased ISTI and the development of the CI.*

There is a strong correlation between the OSTI and the development of CI. The OSTI actually reflects the achievements of STI, and the carrier needed for the development of CI comes from it. New materials, equipment, facilities and technologies produced by STI can continuously generate new cultures, so it is only the OSTI that has achieved breakthroughs. Therefore, the expression form of the CI may undergo subversive changes and make great progress, so that the CI enters a different stage of development. Thus, there is a strong degree of correlation between the two.

Based on the above analysis, a fourth hypothesis is proposed:

*H4*: *There is a strong correlation between the OSTI and the development of the CI.*

The improvement of the STIB is linked to the development of the CI. Both technological innovation and CI are essential carriers of enterprises, and the ultimate purpose of enterprises is to pursue benefits. Thus, the social subjects of the two are similar in their development purposes, which enhances the degree of correlation between the two to some extent. In addition, the economic STIB also push forward the development of CI, which is specifically reflected in that STI provides financial support and consumption means for the development of CI [4]. On the one hand, as the primary productive force, S&T can create greater benefits to boost the development of CI. On the other hand, the promotion of S&T makes the means of cultural consumption diversified and convenient, and the consumption generated by culture eventually becomes part of the STIB.

Based on the above analysis, a fifth hypothesis is proposed:

*H5*: *The improvement of the STIB is related to the development of the CI.*

## 4. Construction of index system for evaluating CD between CI and STI

### 4.1 Selection of the CI indicators

CI, also known as creative industry in western countries, is an emerging industry. Due to the various definitions of CI in different countries, so, the corresponding evaluation systems also have certain differences. Developed European countries mainly use various creative index systems to measure the level and competitiveness of the creative ability of a country or region, such as the 3T Creative Index by Florida [29,30] and the European Creative Index by Florida and Tinagli [29]. Bobirca and Draghici [31] evaluated the creativity of European countries by utilizing the European Creativity Index. Kloudova and Chwaszcz [32] integrated the previous research experience and unified the indicators in the 3T Creativity Index and the European Creativity Index. Petrikova et al. [33] analyzed creative ability based on 3T Creativity Index. Chinese scholars are more likely to construct an index system based on Porter's diamond model, so as to compare and study CIs in different countries [34].

Based on these researches, as well as the Development Index of CI of Provinces and Cities in China 2012 (published by Renmin University of China and The Department of Industry, Ministry of Culture) and China CI Quality Development Index 2019 (jointly developed by the Cultural Economy Research Institute of Central University of Finance and Economics, Beijing Cultural Investment Big Data Co., LTD., Xinhua Net Co., LTD.), an evaluation index system of the high-quality development of CI is built from four dimensions that are industrial innovation (II), industry resources (IR), industry opening (IO) and industry sharing (IS), as Table 1 shows.

In terms of II, cultural innovation provides a driving force for the high-quality development of CI [47]. Innovation is regarded as a measurement indicator of the development of CI, which has been recognized by many scholars. In particular, innovation resources are the basis for the innovation of CI. The number of innovation subjects, innovation consciousness and the accumulated knowledge, technology and information storage are an important reflection of the II ability of a region's CI [48], while innovation performance represents the development results of CI. In terms of specific measurement indicators, CI, as a creative industry, the high-quality and high-level industrial talents will undoubtedly provide a vital talent guarantee for the industrial development [35]. Therefore, the number of personnel in related industries is an

**Table 1. Development index indicators system of CI.**

| First-level indicators | Second-level indicators | Third-level indicators | Literature basis |
|---|---|---|---|
| II | Innovation resources | Number of employees in CI and cultural relics industry | Ren & Xu [35] |
| | | Number of culture, sports and entertainment legal entities | Wang [36], Qin & Cong [37] |
| | | Number of persons engaged in literature and art and scientific research | Florida [29] |
| | Innovation performance | Added value of culture and its related industries (100 million yuan) | Zhu [38] |
| IR | Cultural resources | Number of collections of cultural relics (pieces/sets) | Wang & Zhao [39] |
| | | Number of A-level tourist attractions by region | Cheng & Huang [40] |
| | | Number of cultural relics institutions by category | Fan [41] |
| | | Number of public cultural facilities (public libraries + mass cultural Institutions) | Wang [36] |
| | Policy support | State financial expenditure on culture, sports and media (100 million yuan) | Chen |
| | | Proportion of cultural expenditure in total financial expenditure (%) | Yang & Wang [42] |
| IO | Cross-border cultural and tourism industry development | Inbound overnight visitors (10 thousand people) | Li [43],Wang [42] |
| | | Number of foreign direct investment projects signed in culture, sports and entertainment industries | Wang & Gao [44] |
| | | Amount of foreign direct investment actually utilized in culture, sports and entertainment industry (US $10,000) | Li [43] |
| | Cultural communication | Total number of overseas Chinese cultural centers | Xiang & Li [45] |
| IS | Access to cultural products and services | Public library floor area per 10,000 people ($m^2$) | Wang [36] |
| | | Total public library collection (10,000 volumes) | Wang [36] |
| | | Cultural activities organized by mass cultural institutions (Ten thousand times) | Duan [46] |

important basis for measurement [30]. Besides, the number of relevant institutions also reflects the investment of social resources from the side. Hence, the number of employees in culture industry and cultural relics industry, the number of culture, sports and entertainment legal entities [37,49], and the number of employees engaged in literature and art and scientific research are included in the index system. The added value of culture and its related industries represents the development status of CI from the market level and can be used as evidence to measure the output of the industry [38].

In terms of IR, the development of CI requires a strong cultural resource endowment as the foundation, otherwise CI will become a rootless tree. The cultural resource endowment is not only in need of historical and cultural resources, but also in need of actual cultural resources [46] and assets [50], such as human natural heritage, cultural heritage, infrastructure construction, etc. Therefore, considering the resources of CI, the number of local cultural heritages [39], the number of cultural relics, the number of A-level tourist attractions in each region [40], the number of various cultural institutions of relics [41], and the number of public cultural facilities [36] have become important indicators. In addition to cultural resources, cultural policies [50] are also extremely crucial for industrial development. In the early stage of CI development, it needs to rely on the exogenous power of policy promotion and environmental optimization (Chen, 2016). Hence, government policy support has also become an important driving force for the development of CI. The government can provide subsidies and support to cultural and related industries through cultural policies, formulate relevant laws and regulations and market systems, and provide a sound environment for the development of CI, which can have a direct impact on the development of CI [42]. The amount of government investment in CI can reflect the support of relevant government policies from the side. Therefore, this research referred to the indicators of previous research and used the proportion of

national financial expenditure on culture, sports and media and cultural undertakings in the total financial expenditure as the measurement standard.

Openness is of great significance to the productivity of CI [51], so IO is the only way for the high-quality development of CI. As China further strengthens foreign cooperation and exchanges, the openness of CI has become an important standard to measure the development of CI in China. As a highly-correlated industry, tourism plays a huge role in stimulating and driving the growth of CI [43]. Therefore, this research adopted the development of a cross-border cultural and tourism industry as the measurement standard of IO, and the number of inbound tourists reflects the international competitiveness of the regional tourism industry [36], so inbound overnight tourists were taken as a three-level indicator. At the same time, the amount of foreign direct investments [44] and the actual use of foreign direct investment [43] in culture, sports and entertainment industries both reflect the integration situation of China's culture industry into the global industrial chain, value chain, supply chain, technique and capital chain. Hence, it is also included in the measurement indicators of the development of the cross-border cultural tourism industry. In addition, close cultural exchanges have promoted the mutual acceptance and integration of different ethnic cultures, laying an important foundation for future cooperation. By the means such as setting up overseas cultural centers, the government vigorously propagandises content with strong cultural characteristics at the national level, which plays a very important role in highlighting national characteristics, expanding cultural influence and promoting cultural value identification [45]. Therefore, this research took the total number of overseas Chinese cultural centers as an index of cultural exchange.

In terms of IS, sharing is the end result of the high-quality development of CI, which necessitates the resource integration and supply chain sharing of CI, as well as the sharing of cultural products and services by people all over the country or even the world [2]. As a new pillar industry, the sharing of CI can be measured by the access to cultural products and services. For the selection of specific indicators, the present research included the measurement standards of the floor area of public libraries per 10,000 people, the total collection of public libraries [36] and the number of cultural activities organized by mass cultural institutions into the index system [46].

## 4.2 Selection of the STI indicators

STI is a hot topic in academic research. The evaluation system of STI can be traced back to the 1850s, when the United States systematically assessed its own SI competitiveness. Until the publication of Science Indicators in 1972, the evaluation index system of STI ability was formally established. The evaluation of STI has been attached great importance to at present. Among the authoritative systems constructed, there are the Global Innovation Index (GII) and Knowledge Economy Index (KEI) issued by the World Bank, Comprehensive Innovation Index (SII) and science by the European Union, as well as Technology statistics by Japan. In China, there are National Innovation Index, China Innovation Index (CII), Strategic Emerging Industries Classification (2018), etc. These index systems are of important reference significance for the index construction for this study. There are some differences in the Index systems of STI, but there shows convergence in the general direction during the recent years. For example, although the index system of GII is slightly different from that established in China, they are both constructed from the same four aspects that are the STIE, ISTI, OSTI and STIB. This first-level classification is also used in the construction of the index system in this study, and extensive reference is made to previous index systems to build the second-level index and to construct the evaluation system for STI in this study, as Table 2 shows.

**Table 2. Development index indicators system of STI.**

| First-level indicators | Second-level indicators | Literature basis |
|---|---|---|
| STIE | Number of scientific research and development institutions | Li et al. [52] |
| | Number of postgraduate training institutions in regular universities | Li et al. [52], Zhao [53], Zhao & Huang [54] |
| | Number of popular science activities (times) | Guan et al. [55], Feng et al. [56], Wu & Yang [57] |
| | Number of participants in popular science activities (ten thousand persons) | Guan et al. [55], Feng et al. [56],Wu & Yang [57] |
| | Number of science museums | Chittenden & Commentary [58], Liu & Ma [59] |
| ISTI | Research and experimental development expenditure (100 million yuan) | Heffernan [60], Díaz-Díaz & De Saá-Pérez [61] |
| | Number of employees engaged in research and development works in relevant institutions (ten thousand persons) | Jv & Liu [62] |
| OSTI | Number of domestic patent applications authorized (pieces) | Gao et al. [4] |
| | Number of published SI papers (ten thousand) | Wu & Hu [63] |
| | Number of published SI works (species) | Wang [36] |
| STIB | Turnover of technology market (100 million yuan) | Negroponte, Liu & Chen [64] |
| | Sales revenue of new products of high-tech industry (ten thousand yuan) | Li & Xu [65] |

In terms of STIE, innovation subjects include the government, enterprises and universities [66]. Scientific research institutions established by the government and enterprises provide research talents with places for research activities and sources of funds. As an important research subject, universities have played an important role in STI [54], and they have also provided research sites for numerous scientific research personnel and media. Especially in recent years, along with the increase of input into higher educations, universities have acted as an increasingly important role in the national STI system [53]. Graduates are important STI subjects. Accordingly, the number of STI institutions and the number of graduate cultivating institutions and universities are taken as the secondary indicators. Meantime, the popular science activities (PSAs) [55–57] and science and technology museums (STMs) [58,59] are important ways at a social level to raise the focus of the general public on STI and the STI enlightenment of the youth. The times and participants of PSAs, and the number of STMs were introduced into the secondary index system.

In terms of ISTI, ISTI is mainly reflected in two aspects: investment of STI capital and input of STI talents [67]. Specifically, the R&D investment, as the material basis of STI, is a key factor for improving a country's innovation capability [60], which can also prove the government's emphasis on STI, and the scale of the government's ISTI can significantly promote the scale of OSTI. For example, the increase in the proportion of R&D input can promote the output of new products [61]. Hence, R&D input is taken as a secondary-level indicator. In terms of STI talents, R&D personnel are the most important subjects of STI, and the number of R&D personnel directly represents the investment amount of STI talents. therefore, the number of R&D personnel in scientific R&D institutions is considered as a secondary-level indicator.

OSTI refers to various forms of intermediate results generated by STI activities, which is the embodiment of the level and ability of STI [68]. Currently, it is recognized that papers and patents are effective ways to measure the OSTI of a country [63]. In addition, it is found that scientific books are also an important output result of STI, which is very easy to be ignored by scholars. Books, as an important transmission carrier of STI achievements, play a decisive role

in popularizing the knowledge of science and technology, transferring information, and promoting the S&T progress of science and technology [69]. Accordingly, in the present study, the number of domestic patent application authorization, published scientific papers and published scientific works were included in the evaluation system as indicators to measure the OSTI.

In terms of STIB, the turnover of the technology market can directly show the benefits brought by the transformation of STI, so it is widely applied as an important indicator [7,64]. While the sales revenue of new products in the high-tech industry [65] can reflect the marketization ability of a country's STI achievements. Therefore, the transaction volume of the technology market and sales revenue of new products in the high-tech industry were included in the measurement index of STIB.

## 5. Methodology

After the construction of the evaluation index system, the comprehensive level of China's CI and STI subsystem was obtained by using the entropy value method (EVM). Then, the coupling coordination degree (CCD) model was used to measure the coupling relationship between the two systems of CI and STI in China at the macro level, and the influence of each indicator of STI on the development of CI was investigated in detail at the micro level by using the grey correlation degree method (GCDM). CCD and GCDM are used to investigate the macro and micro relations between the two systems. This method has been used by Chu et al. [70] to study the coordinated development of logistics industry and financial industry. More studies are used in environmental protection and sustainable development direction [71] or logistics industry development direction [72].

### 5.1 Model Setting

**5.1.1 Entropy Value Method (EVM).**   Entropy is a physical unit of measurement. The entropy rule is a research method to determine weights based on the information provided by entropy. When there are multiple indicators in an evaluation system, it is difficult to integrate them due to different dimensions and orders of magnitude. The weight analysis of indicators utilizing EVM can effectively avoid the randomness of the subjective weight and the limitation of the objective weight, solve the problems of information repetition among variables and improve the scientificity and accuracy of evaluation [73]. In this study, the original data of each year was standardized before analysis, and EVM was used to objectively weigh each indicator to obtain the weight matrix. Finally, the development index of CI was obtained by multiplying the weight of each indicator by the standardized value and adding up these multiplied values. The specific method is as follows.

(1) Standardization process of original data

Since all the measures adopted in this study are positive indicators, only positive indicators are used in the original data processing, and non-dimensional treatment are adopted in this study. In other words, an evaluation index in a certain year is $X_{ij}$, its standardized value is $Y_{ij}$, and the positive index is:

$$Y_{ij} = \frac{X_{ij} - \min(X_{ij})}{\max(X_{ij}) - \min(X_{ij})}, i = 2011, 2012, \ldots, 2020, j = 1, 2, \ldots, n \qquad (1-1)$$

(2) EVM to determine the weight of indicators

The specific proportion $P_{ij}$ of the $i^{\text{th}}$ evaluation object under item $j$ is calculated as follows:

$$P_{ij} = \frac{X_{ij}}{\sum\limits_{i=1}^{m} X_{ij}}, i = 2013, 2014, \ldots, 2021, j = 1, 2, \ldots, n \tag{1-2}$$

The entropy value $E_j$ of the $j^{\text{th}}$ evaluation indicator is calculated as follows:

$$E_j = -\frac{1}{\ln m} \sum_{i=1}^{m} P_{ij} \ln(P_{ij}), j = 1, 2, \ldots, n \tag{1-3}$$

The weight $w_j$ of the $j^{\text{th}}$ evaluation index is calculated as:

$$\omega_j = \frac{1 - E_j}{\sum\limits_{j=1}^{n}(1 - E_j)}, j = 1, 2, \ldots, n \tag{1-4}$$

(3) Calculation of the subsystem index score

$$Z_i = \sum_{j=1}^{m} \omega_j Y_{ij}, j = 1, 2, \ldots, n \tag{1-5}$$

After calculating the original data of the CI and STI subsystems according to the above weights, the entropy score can be obtained, and the score calculated in above process is between 0 and 1, and the score is the total score of the development status of the cultural industry and scientific and technological innovation, the score characterizing the development condition of cultural industry and science and technology innovation in China.

**5.1.2 CCD model.** The word, "coupling", comes from physics and can be used to analyze the dynamic relationship between multiple interrelated systems. The coupling degree reflects the degree of interdependence and mutual limitation of multiple systems. The coordination degree measures the benign coupling degree among the coupling relations of multiple systems and reflects the quality of coordination [52]. In this research, it can be used to measure the interdependence, coordination and promotion relationship between CI and STI. Since CI and STI are interdependent and develop coordinately, their coupling will also increase, otherwise, it will decrease.

(1) Overall comprehensive contribution model

We set $W_i$ and $K_i$ as the weight of the $i^{th}$ index in the subsystem of CI and STI, respectively, and calculate the weight of each index by the entropy weighting method. The overall comprehensive contribution models of indicators in the subsystem of CI and STI are expressed as follows:

$$U = \sum_{i=1}^{m} W_i \times U_i, G = \sum_{i=1}^{m} K_i \times G_i \tag{2-1}$$

(2) Calculation of coupling degree

**Table 3. Discriminant standard of coupling degree.**

| Coupling phase | Low-level coupling stage | Rivalry stage | Running-in stage | High-level coupling stage |
|---|---|---|---|---|
| Coupling degree C | (0,0.3] | (0.3,0.5] | (0.5,0.8] | (0.8,1] |

The coupling degree between CI and STI can be calculated by the following formula:

$$C = \frac{2\sqrt{U \times G}}{U + G} \tag{2-2}$$

where, C represents the degree to which CI and the STI system influence each other through their respective coupling factors. According to the coupling degree, CI and STI can be divided into four coupling stage [74], as shown in Table 3.

(3) Results of CCD

CCD of CI and STI can be calculated by the following formula:

$$D = \sqrt{C \times T}, T = aU + bG \tag{2-3}$$

where, $D$ represents the CCD; $T$ represents the comprehensive evaluation index reflecting the overall synergistic effect of CI and STI; a and b represent the importance degree of the two subsystems of CI and STI, respectively. It is assumed that the two systems are equally important, so, a = b = 0.5. According to the existing literature [75], the degree of coordination can be divided into 10 levels, as shown in Table 4.

**5.1.3 GCDM.** Considering the complexity, relevance, interlacing and timing of the coupling effect between CI and STI, GCD model (GCDM) will be used in this study to conduct further analysis from the micro perspective and investigate in detail the influence of each indicator of the STI system on CI, exploring interaction and revealing the coupling mechanism between the two systems.

GCDM is a measurement method to describe the strength, size and order of relationships between factors of things or systems, which was firstly put forward by Deng [76]. GCDM can be used for quantitative analysis for the dynamic development process of a system to examine whether the factors in the system are closely connected, so as to identify the status of primary and secondary factors affecting the development of the system and measure the correlation degree between sequences [76]. Since GCDM has no high requirement on sample size, and does not require typical distribution rules during data analysis, and the analysis process

**Table 4. Classification standard of CCD.**

| D-value interval of CCD | Coordination level | CCD |
|---|---|---|
| (0.0,0.1) | 1 | Extreme maladjustment |
| [0.1,0.2) | 2 | Serious maladjustment |
| [0.2,0.3) | 3 | Moderate maladjustment |
| [0.3,0.4) | 4 | Mild maladjustment |
| [0.4,0.5) | 5 | On the verge of maladjustment |
| [0.5,0.6) | 6 | Barely coordination |
| [0.6,0.7) | 7 | Primary coordination |
| [0.7,0.8) | 8 | Intermediate coordination |
| [0.8,0.9) | 9 | Good coordination |
| [0.9,1.0) | 10 | High-quality coordination |

combines quantitative and qualitative analysis, so, GCDM has extensive practicability and is widely used in many fields [77].

(1) Dimensionless processing

In order to ensure the comparability of all indicators, guarantee the scientific nature of this study, and eliminate the different influences due to different dimensions of each indicator, standardization processing method are adopted for the obtained initial data of CI and STI.

Specifically, the initial value method is used for standardization processing, and the specific formula is as follows:

$$X'_j(i) = \frac{X_j(i)}{X_j(1)} \tag{3-1}$$

$$Y'_j(i) = \frac{Y_j(i)}{Y_j(1)} \tag{3-2}$$

where, $X'_j(i)$, $Y'_i(i)$ is the initialized value of the $j^{th}$ indicator in the $i^{th}$ year, $j = 1,2\ldots, n$; $i = 1,2,3,4,5,6,7,8,9,10$; $X_j(i)$, $Y_j(i)$ is the actual value of the $j^{th}$ index in the $i^{th}$ year.

(2) Grey Correlation Coefficient (GCC) and GCD

① Evaluate the sequences of different
The formula for the evaluate the sequences of different is:

$$\Delta_j(i) = |X'_j(i) - Y'_j(i)| \tag{3-3}$$

Before calculation, we conduct dimensionless processing for the data of CI and STI, and then obtain the evaluation evaluate the sequences of different each indicator according to Formula (3-3). After that, we calculate the two-step difference, with the formula being:

$$M = \max \Delta_j(i), \ m = \min \Delta_j(i) \tag{3-4}$$

② Calculate GCC
The calculation formula of the GCC is:

$$\xi(i) = \frac{m + pM}{\Delta_j(i) + pM} \tag{3-5}$$

With four evaluation items (STIE, ISTI, OSTI and STIB), we conduct GCDM. The entropy value (EV) of the CI is taken as the "reference value" (master sequence), to investigate the GCD of the four evaluation items (STIE, ISTI, OSTI and STIB) and the EV. In this study, we calculate the correlation with the distinguish coefficient as 0.50.

③Calculate GCD
We obtain the CDV between CI and STI by weighted approach with the above correlation coefficient results. Then we use the CDV to evaluate and rank the 10 assessment objects. According to the correlation value, the CDV can be calculated for evaluation. The calculation formula of the GCD is as follows:

$$R_j = \frac{1}{10} \sum_{i=1}^{10} \xi_j(i) \tag{3-6}$$

Where, $R_j$ represents the GCD value between the $i^{th}$ indicator and CI.

**Table 5. Classification standard of CCD.**

| Correlation degree range | Degree of correlation |
|---|---|
| (0.0,0.35) | Weak correlation |
| [0.35,0.65) | Moderate correlation |
| [0.65,1) | Strong correlation |

According to Formula (3-6), we calculate the GCD between CI and STI. Then referring to the existing literature, we divide the GCD into three grades, as shown in Table 5.

## 5.2 Data sources

The data are from the National Bureau of Statistics of the People's Republic of China (2013–2021), the Statistical Yearbook of Chinese Culture and Related Industries (2013–2021), the Statistical Yearbook of China (2013–2021), the Statistical Yearbook of China Science and Technology (2013–2021).

## 6. Analysis and conclusions

### 6.1 Calculation of Weight based on EVM

The weight of each indicator of CI and STI was calculated based on the EVM, and the weight of each indicator is shown in Tables 6 and 7 respectively.

As can be seen from Table 6, among the three levels of indicators of CI, the weight of IR in the first-level index is the largest, reaching 29.51%. In the secondary indicators, the weight of access opportunities for cultural products and services in CI is 25.49%, followed by policy support of 19.75%. Among the three levels of indicators, the two largest proportions are the proportion of cultural spending in total financial expenditure and the added value of culture and its related industries. It can be seen that, for CI, the cost of cultural undertakings is the main factor affecting its development, and the added value of culture and its related industries also has a significant impact on it, which indicates that there is a close relationship between CI and other industries.

As can be seen from Table 7, among the first-level indicators of STI, STIB accounts for the largest proportion, which is 27.9%. Among the secondary indicators, the highest weight is technology market turnover and research and development spending. Therefore, technological innovation, technological market turnover, and research and development expenditures are the main factors affecting its development.

### 6.2 Analysis of coupling coordination between CI and STI

According to the calculation formula of Coupling degree (2–2) and CCD (2–3), the annual comprehensive contribution value U of CI, comprehensive contribution value G of STI, coupling correlation degree C and CCD D can be obtained, as shown in Tables 3 and 4.

In order to further explore the evolution logic of coupling correlation degree and coordination degree between CI and STI, a comparative analysis was carried out by integrating coupling degree and CCD, with the results shown in Fig 1.

It can be found that from Table 8 and Fig 1:

(1) Overall, the coupling degree is above 0.8, basically at a relatively high level in the coupling stage. This conclusion is consistent with the measurement results of qualitative or alternative models used by many scholars (Day, 2008; Ding, 2018), STI also plays a supporting role in the development process of CI, but it can also be found in numerous examples that CI is

**Table 6. Weight of each indicator of CI.**

| First-level indicators | Indicator weight | Second-level indicators | Indicator weight | Third-level indicators | Index weight |
|---|---|---|---|---|---|
| II | 24.58% | Innovation resources | 12.78% | Number of employees in CI and cultural relics industry (persons) | 6.11% |
| | | | | Number of cultural, sports and entertainment legal entities (units) | 4.05% |
| | | | | Number of persons employed in literature, art and scientific research | 2.61% |
| | | Innovation performance | 11.8% | Added value of culture and its related industries (100 million yuan) | 11.8% |
| IR | 29.51% | Cultural resources | 9.76% | Number of collections of cultural relics (pieces/sets) | 2.33% |
| | | | | Number of A-level tourist attractions by region (number) | 3.17% |
| | | | | Number of cultural relics institutions by category(units) | 1.79% |
| | | | | Number of public cultural facilities (public libraries + mass cultural Institutions) (units) | 2.45% |
| | | Policy support | 19.75% | State financial expenditure on culture, sports and media (100 million yuan) | 6.32% |
| | | | | Proportion of cultural expenditure in total financial expenditure (%) | 13.43% |
| IO | 20.42% | Development of the cross-border cultural and tourism industry | 13.28% | Inbound overnight visitors (10 thousand people) | 1.77% |
| | | | | Number of foreign direct investment projects signed in culture, sports and entertainment industries (units) | 8.55% |
| | | | | Amount of foreign direct investment actually utilized in culture, sports and entertainment industries (US $10,000) | 3% |
| | | Cultural communication | 7.14% | Total number of overseas Chinese cultural centers (units) | 7.14% |
| IS | 25.49% | Access to cultural products and services | 25.49% | Public library floor area per 10,000 people (m$^2$) | 7.49% |
| | | | | Total public library collection (10,000 volumes) | 10.54% |
| | | | | Cultural activities organized by mass cultural institutions (Ten thousand times) | 7.45% |
| Total | 100% | | 100% | | 100% |

**Table 7. Weight of each indicator of STI.**

| First-level indicators | Index weight | Second-level indicators | Index weight |
|---|---|---|---|
| STIE | 24.93% | Number of scientific research and development institutions | 3.38% |
| | | Number of postgraduate training institutions in regular universities | 2.77% |
| | | Number of popular science activities (times) | 3.32% |
| | | Number of participants in popular science activities (ten thousand) | 10.07% |
| | | Number of science museums | 5.39% |
| ISTI | 22.10% | Research and experimental development expenditure (100 million yuan) | 12.40% |
| | | Number of employees engaged in research and development works in relevant institutions (ten thousand persons) | 9.70% |
| OSTI | 25.07% | Number of domestic patent applications authorized | 8.39% |
| | | Number of published SI papers (ten thousand) | 10.01% |
| | | Number of published SI works | 6.67% |
| STIB | 27.90% | Turnover of technology market (100 million yuan) | 17.61% |
| | | Sales revenue of new products of high-tech industry (ten thousand yuan) | 10.29% |
| Total | 100% | | 100% |

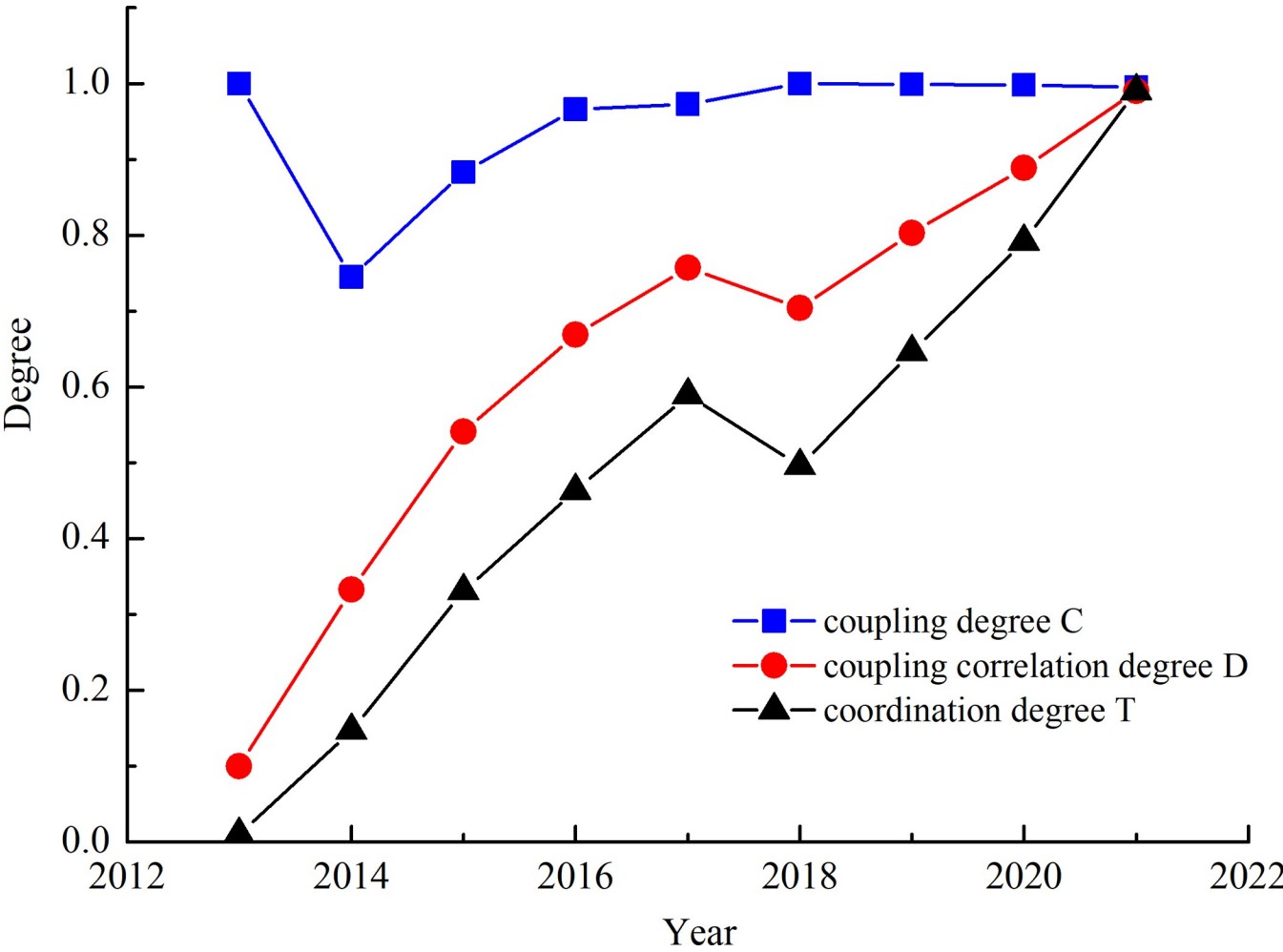

**Fig 1. Coupling degree sequence of CI and STI.**

**Table 8. CCD of CI and STI in China.**

| Year | Coupling degree C value | Coupling phase | Coordinate index T-value | CCD D-value | CL | CCD |
|------|------|------|------|------|------|------|
| 2021 | 1.000 | High-level coupling phase | 0.990 | 0.995 | 10 | High-quality coordination |
| 2020 | 0.998 | High-level coupling phase | 0.792 | 0.889 | 9 | Good coordination |
| 2019 | 0.999 | High-level coupling phase | 0.646 | 0.803 | 9 | Good coordination |
| 2018 | 1.000 | High-level coupling phase | 0.496 | 0.704 | 8 | Intermediate coordinate |
| 2017 | 0.973 | High-level coupling phase | 0.589 | 0.757 | 8 | Intermediate coordinate |
| 2016 | 0.966 | High-level coupling phase | 0.463 | 0.669 | 7 | Primary coordination |
| 2015 | 0.883 | High-level coupling phase | 0.331 | 0.541 | 6 | Barely coordination |
| 2014 | 0.754 | Running-in stage | 0.147 | 0.333 | 4 | Mild maladjustment |
| 2013 | 1.000 | High-level coupling phase | 0.010 | 0.100 | 2 | Serious maladjustment |

(Note: Data from 2013 to 2021).

also pushing the innovation and progress of S&T. For example, the prevailing digital technologies such as multimedia, VR and others have become the carriers of cultural creativity, all for the purpose of satisfying the new technologies spawned by the rapid development of the CI. At this level, it is the CI that gives full play to its advantages of innovation and diffusion, making traditional STI get rid of the shackles of rigid thinking and providing new development directions and ideas for it. In other words, as a creative industry, the CI needs the achievements of STI as a material carrier to promote its development. Correspondingly, the CI can push back the progress of STI by providing spiritual motivation and other ways. Therefore, CI and STI have a deeply coupling effect.

(2) With only 2014 in the running-in stage. According to the index values of subsystems, compared with the previous year, the growth rate of the STI index in 2014 is significantly slower than that of the CI index, and the sub-indexes of STIE, ISTI, OSTI and STIB even regresses compared with 2013, which is the main cause for the low coupling degree of the two. Referring to the, related information, it can be found that compared with 2013, the patent authorization number index of every 10 thousand R&D personnel in 2014 is decreased by 6.3%; the proportion of high-tech products exports in goods exports is decreased by 5.7% (State Statistics Bureau: The 2014 China Innovation Index was 158.2.) over the previous year. All of these, to some extent, have effect on the development of China's STI in 2014.

(3) CCD is generally on the rise, so the first hypothesis should be accepted. This point has also been confirmed in the studies of many scholars [78] (Han, 2019; Su & Kang, 2022). Specifically, the CCD in this study can be divided into three stages: the CCD between China's CI and STI is between 0.1–0.4 during 2013–2014, indicating that China's CI and STI are in a disequilibrium stage; from 2015 to 2018, the CCD between China's CI and STI is between 0.4 and 0.6, indicating that China's CI and STI are in the initial and intermediate coordination stage. From 2019 to 2021, the CCD between China's CI and STI is between 0.8–1, and it turns from good coordination to high-quality coordination, indicating a strong coupling relationship between the two. The development process of coordination between the two accords with the early development of S&T and the slow development of the CI in our country, which resulted in a large gap and low coordination degree, but the rapid development of the CI in the late period, resulting in the increased coordination degree of CI and STI. Overall, the degree of coordinated development between the two has shown an upward trend, indicating that the deep-seated relationship between CI and STI has been discovered by scholars and policy makers, who have promoted their integration, symbiosis and coordinated development through various means.

(4) During the three stages, the CCD declines to a certain extent in 2018, which is caused by the decrease in the comprehensive score of the CI in that year compared with the previous year. The growth rate of CI slows down in 2018. For the possible reasons, on the one hand, the domestic CI market becomes more mature, the capital market of CI tends to be rational, and the phenomenon of investment and financing subjects blindly following the trend diminishes [79]. On the other hand, it is also directly related to the overall slowdown of the country's economic growth rate and the "new normal" environment of economic development [80].

## 6.3 Analysis of key factors on the coupling effect of CI and STI

With the 4 evaluation items of STI (STIE, ISTI, STIO and STIB) from 2013–2021, GCD analysis was conducted, and the overall comprehensive contribution score of CI was taken as the

**Table 9. Correlation coefficient results.**

| Year | STIE | ISTI | STIO | STIB |
|---|---|---|---|---|
| 2021 | 1.000 | 1.000 | 1.000 | 1.000 |
| 2020 | 0.467 | 0.743 | 0.771 | 0.639 |
| 2019 | 0.729 | 0.704 | 0.864 | 0.634 |
| 2018 | 0.914 | 0.942 | 0.776 | 0.659 |
| 2017 | 0.576 | 0.385 | 0.628 | 0.333 |
| 2016 | 0.696 | 0.403 | 0.608 | 0.349 |
| 2015 | 0.627 | 0.401 | 0.357 | 0.349 |
| 2014 | 0.784 | 0.528 | 0.416 | 0.467 |
| 2013 | 0.607 | 0.666 | 0.919 | 0.666 |

"reference value" (parent series). In this paper, we studied the correlation between the four evaluation items (STIE, ISTI, STIO and STIB) and the overall contribution score of CI (correlation degree). The resolution coefficient was set to 0.50, and then correlation degree value (CDV) was calculated according to the correlation coefficient calculation formula to calculate the correlation value for evaluation. With GCD analysis, the correlation coefficient results of CI and STI were obtained, as shown in Table 9.

The results of the above correlation coefficients were weighted to obtain the CDV. The CDV was used to evaluate and sort the data of 9 years, and Table 10 was obtained. It can be seen from the table that the CDV is between 0 and 1. The larger the value is, the stronger the correlation is with the overall comprehensive contribution score (parent sequence) of the CI, that is, the higher the evaluation is.

According to the above calculation results shown in Table 10, for the four evaluation items, the GCD between all indicators and the development of CI is higher than the measurement value of 0.35. The four evaluation items in the index of STI all have a certain correlation degree with the CI, so the remaining four hypotheses should be accepted. Among them, the comprehensive evaluation of STIE is the highest (the correlation degree: 0.711), followed by STIO (the correlation degree: 0.704), and the correlations of the two indicators with CI are higher than 0.65, which shows strong correlation, indicates that the two STI indicators are closely correlated with CI. The correlation degree of STIB ranks the last, being only 0.566, showing moderate correlation, indicates that the correlation between STIB and CI is relatively small.

## 7. Discussion

This study constructed an evaluation system for CI and STI and found that the CCD between the two systems gradually increased, revealing the relationship between CI and STI at a macroscopic level. In this work, we further analyze the degree of correlation between specific STI indices and CI and reveal the microscopic mechanism behind this coupling phenomenon from the index level. This study will help accurately assess the development of CI and STI,

**Table 10. Results of correlation between STI and CI.**

| Hypothesis | Evaluation of item | Correlation | Rank |
|---|---|---|---|
| H2: STIE→CI | STIE | 0.711 | 1 |
| H3: ISTI→CI | ISTI | 0.641 | 3 |
| H4: OSTI→CI | OSTI | 0.704 | 2 |
| H5: STIB→CI | STIB | 0.566 | 4 |

serve as a reference for further promoting the coordinated development of CI and STI in China, and also provide Chinese experience for the coordinated development of CI and STI in the world.

The advantages of this study are twofold. First, the index system for measuring CI and STI has not been unified at home and abroad so far. However, this study has established a relatively complete indexing system by fully and extensively reviewing the literature, which can serve as an experience and reference for future research. Second, this study adopts a quantitative research approach and empirically investigates the problem from both macro and micro perspectives. To the best of our knowledge, this is the first empirical study to analyze CI and STI from both macro and micro perspectives. Moreover, this study does not use traditional econometrics, but the EVM method, which is also innovative in its approach.

The limitation of this study is that the GCDM adopted can only reflect the degree of correlation between CI and STI, but does not further reveal whether there is a causal relationship between the two, namely: Whether the development of the CI and STI can promote the development of the other or not, this part of the study can be further discussed by Granger test, so as to explore the more specific and detailed endogenous relationship between the CI and STI. This is what the author expects to do next.

## 8. Conclusions and recommendations

### 8.1 Conclusions

This study constructed an index system reflecting CI and STI, empirically analyzed the coupling coordination effect of the two by utilizing the entropy method and CCD model, and further analyzed the key factors of interaction between the two by using GCD. The study results indicate that:

(1) For CI, IR, policy support and cultural undertakings cost accounted for the highest weights of the first, second and third indicators, respectively, indicating that for endogenous links, the three were the main factors affecting its development;

(2) During the nine years from 2013 to 2021, the CCD between China's CI and STI was basically at a high level, being relatively stable;

(3) The CCD of China's CI and STI gradually turned to high-quality coordination from the initial moderate imbalance, showing an upward trend on the whole, indicating that China has gradually embarked on a path of coordinated development and benign interaction between CI and STI;

(4) Among the indicators of STI, the average correlation degree of the overall contribution scores of STIE and STIO to CI was higher than 0.7, which is a strong correlation, indicating that these two factors are the key factors restricting the development of China's CI.

### 8.2 Recommendations

Based on the theoretical and empirical analysis, this research makes the following suggestions for the coordinated development of CI and STI:

(1) Increase spending on industrial resources and cultural programs and increase policy support. Industrial resources, policy support and cultural spending have played a decisive role in the development of the cultural industry. Therefore, we need to strengthen the guidance and support at the national level, combine the "visible hand" with the "invisible hand", give

play to the leading role of macro-control on the basis of respecting the law of market economy, and support the development of the cultural industry with active industrial policies. On the one hand, the state can introduce relevant policies and measures to provide favorable treatment in terms of capital and taxes to cultural industry enterprises, increase capital investment, facilitate development, and provide development guarantees from the system, according to market demand and development trends. On the other hand, control and supervision should be strengthened in response to the endless cultural infringement incidents in recent years.

(2) To create a good environment for STI and influence the development of the CI to better and faster development. Encourage cultural and creative subjects (including cultural and creative enterprises and individuals) to carry out content and scene innovation. Fully excavating Chinese traditional culture and literary creation in line with the contemporary development trend, combine the traditional (historical) with The Times, so as to enrich the cultural content with novelty and The Times. We should increase the cultivation of high-level cultural talents, improve the "people-to-people bond" through international exchanges, introduce and absorb foreign excellent cultural forms and contents while going out, and attract outstanding cultural people to participate in cultural exchanges and innovation.

(3) We should attach importance to the influence of STIO on the development of CI, and promote the development of cultural industry by optimizing the carrier. Increase investment in cultural and technological infrastructure, including the construction of buildings, venues, studios and other aspects. The development of CI and STI requires the support of carriers. In the aforementioned research, S&T museums and related S&T activities have become key factors in promoting the coupled development of CI and STI. Therefore, it is necessary to increase investment in related infrastructure to provide material venues for the coupled development of CI and STI. Actively adopt new technologies, including 5G technology and big data technology, to better design, present and market cultural content. Invest in research and development, guide cultural and scientific personnel to cooperate and jointly develop, actively promote scene innovation and use STI to make the CI more attractive.

(4) Vigorously promote STI and promote the rapid development of the CI. The development of STI has a certain synergistic effect on the CI, so we need to seize this main "booster", give full play to the promoting role of STI in the development of CI, implement the strategy of "strong S&T" driving "strong culture", and vigorously develop emerging S&T. Seize the commanding heights of S&T to realize the leapfrog development of our CI, so as to overtake the curve and build a cultural power. For example, we should vigorously develop 5G technology, lay out the meta-universe, and realize the rapid development of the CI by creating new growth poles, developing new business ecosystems and business forms, and improving the transmission of cultural products.

## Supporting information

**S1 Data.**
(XLSX)

## Author Contributions

**Conceptualization:** Zhenni Yu, Jian Yu.

**Data curation:** Zhenni Yu.

**Formal analysis:** Zhenni Yu.

**Investigation:** Zhenni Yu.

**Methodology:** Zhenni Yu.

**Project administration:** Zhenni Yu.

**Resources:** Zhenni Yu.

**Software:** Zhenni Yu.

**Supervision:** Zhenni Yu, Jian Yu.

**Validation:** Zhenni Yu.

**Visualization:** Zhenni Yu.

**Writing – original draft:** Zhenni Yu.

**Writing – review & editing:** Jian Yu.

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
