## [Decision Letter · Decision Letter 0]

23 Jan 2023

PONE-D-22-34472Evaluation of the Coordinated Development between Chinese Cultural Industry and Scientific and Technological InnovationPLOS ONE

Dear Dr. yu,

Thank you for submitting your manuscript to PLOS ONE. After careful consideration, we feel that it has merit but does not fully meet PLOS ONE’s publication criteria as it currently stands. Therefore, we invite you to submit a revised version of the manuscript that addresses the points raised during the review process.

I recommend that it should be revised taking into account the changes requested by the reviewers. Since the requested changes include valuable and constructive reviews, I would like to give you a chance to revise your manuscript. The revised manuscript will undergo the next round of review by same reviewers.

We look forward to receiving your revised manuscript.

Kind regards,

Baogui Xin, Ph.D.

Academic Editor

PLOS ONE

Journal Requirements:

"This paper is supported from Soft Science Research Project of Beijing Science and Technology Plan "Research on Innovative Development Model of Science and Technology promoting Cultural Creative Industry" (#Z09031001590911)"

"This paper is supported from Soft Science Research Project of Beijing Science and Technology Plan "Research on Innovative Development Model of Science and Technology promoting Cultural Creative Industry" (#Z09031001590911) .The funders had specify  role in study design, data collection and analysis."

Reviewers' comments:

Reviewer's Responses to Questions

**Comments to the Author**

1. Is the manuscript technically sound, and do the data support the conclusions?

Reviewer #1: Yes

Reviewer #2: Yes

2. Has the statistical analysis been performed appropriately and rigorously? 

Reviewer #1: No

Reviewer #2: I Don't Know

3. Have the authors made all data underlying the findings in their manuscript fully available?

Reviewer #1: Yes

Reviewer #2: No

4. Is the manuscript presented in an intelligible fashion and written in standard English?

Reviewer #1: No

Reviewer #2: Yes

5. Review Comments to the Author

Reviewer #1: Comments for PONE-D-22-34472

The authors carried out the analysis on the index system of coordinated development between cultural industry (CI) and scientific and technological innovation (STI). The authors did not use the traditional Econometric method, but via EVM method. To be honest, I did not familiar with the method performed in this study. But, I tried my best to learn it and checked them seems right in this manuscript. The most three concerns for this study are: 1) I still strongly recommend authors propose several hypotheses in the literature review or the construction of the index system. I mean that it will be better to test your hypotheses not only just build one index. further, if one index was constructed, the next will be to show its advantage compared with other related indexes. 2) I am confused by figure 1 which will be one of the most results in the present study. Does the maximum value of the Y-axis equal 1? if the maximum is 1, how about the future of the coupling degree because the CD was compared with the adjacent two year. Mostly, a statistical way to show the changes will be better. 3) lack of discussion section. I suggest to add this section to show the advantage of this index compared with others.

specific comment

The most specific comments are the grammar, which should be checked by an English expert. The English is not my first language, but I still found lots errors in the grammar logic. For example, in the first two sentences, CI should be fully written down when it occurrs in the first time.

Because there were no line numbers in the manuscript, the specific comments will be listed as the detailed table and figures.

Table 6: The first-level indicators of II. The indicator weight seems wrong? 12.78% should be 10.7?%?

Figure 1: I believe the figure 1 should be re-draw following the instruction of the PLos ONE.

Reviewer #2: This manuscript attempted to explore the correlation between Chinese cultural industry and scientific and technological innovation. Unlike previous studies that mainly took a qualitative approach, this study attempted a quantitative approach to this relationship. It is also encouraging that the authors presented various literature on the relationship between the cultural industry and science and technology innovation. Exploring the relationship between cultural industries and science and technology innovation in quantitative perspective seems interesting, but challenges involved in estimation must also be taken seriously.

[Major Points]

1. Authors should be careful not to interpret correlations as causal relationships throughout the manuscript. For example, expressions implying a causal relationship, such as “the effect of a specific STI index on CI,” should be avoided and modified to indicate a correlation.

2. The author used EVM, CCD, and GCD for empirical analysis.

However, this manuscript lacks an explanation of whether this analytical process is valid in the context of this analysis.

Authors are requested to provide detailed examples of similar analyzes performed using this methodology.

3. Explain how the authors controlled for the third confounder in the empirical estimate of the correlation.

4. Sufficient evidence needs to be presented to justify the findings of the analysis.

5. In order to secure the reliability of the indicator, it is necessary to describe in more detail other cases where this indicator is used.

6. Between conclusions and recommendations The logical connection between conclusions and recommendations needs to be strengthened. To this end, it is necessary to supplement the recommendations based on the results of analyzing the relationship between CI and STI.

Considering the title and design of this study, it is necessary to make recommendations based on the relationship between the cultural industry and STI, not the cultural industry alone.

[Minor Points]

1. The 'Recommendations' section is described in the order of (1) (4) (5). It needs to be modified to (1) (2) (3)

6. PLOS authors have the option to publish the peer review history of their article (what does this mean?). If published, this will include your full peer review and any attached files.

Reviewer #1: No

Reviewer #2: No

---

## [Author Response · Author response to Decision Letter 0]

9 Mar 2023

Reviewer #1: Comments for PONE-D-22-34472

The authors carried out the analysis on the index system of coordinated development between cultural industry (CI) and scientific and technological innovation (STI). The authors did not use the traditional Econometric method, but via EVM method. To be honest, I did not familiar with the method performed in this study. But, I tried my best to learn it and checked them seems right in this manuscript. The most three concerns for this study are: 1) I still strongly recommend authors propose several hypotheses in the literature review or the construction of the index system. I mean that it will be better to test your hypotheses not only just build one index. further, if one index was constructed, the next will be to show its advantage compared with other related indexes. 2) I am confused by figure 1 which will be one of the most results in the present study. Does the maximum value of the Y-axis equal 1? if the maximum is 1, how about the future of the coupling degree because the CD was compared with the adjacent two year. Mostly, a statistical way to show the changes will be better. 3) lack of discussion section. I suggest to add this section to show the advantage of this index compared with others.

R: 1) Thanks for the good suggestion. According to the reviewer's suggestions, we have added the chapter of hypothesis in the manuscript, and put forward five hypotheses, such as the strong coupling and coordination relationship between cultural industry and scientific and technological innovation. We demonstrate the reasons for this hypothesis and then prove it in the analysis and conclusion of the manuscript. The above discussion is added in the revised manuscript as Section 3, as highlighted by the yellow background.

2) The maximum value on the Y-axis is 1. According to the definition of coupling correlation degree, it can be seen that the upper right of equal sign is the geometric mean, while the lower is the algebraic mean, so the value range of C is [0,1]. Referring to the existing literature, it can also be seen that the result of coupling correlation degree and coupling coordination degree must be between [0,1]. And the future coupling degree will not exceed 1, because the calculation result of the coupling correlation degree model itself is a dynamic correlation trend, and the calculation result is a relative value rather than an absolute value. Therefore, if a more optimized result occurs, the current maximum value will decrease by a corresponding degree. For example, if the calculation year is extended to 2022, even if the coupling correlation degree and coupling coordination degree in 2022 are better than that in 2021, it will be the highest value, namely 1, but the current calculation of 2021 as the highest value "1" will decline.

Statistical methods are not used in the manuscript to show the changes of A, which is because the purpose of the study is to measure the mutual influence and collaboration between the two systems of cultural industry and scientific and technological innovation, rather than the changes of the two systems themselves. If only statistics are used for surveying and mapping, it is difficult to reflect the degree of connection between the two. However, the definition of the reference coupling coordination degree model is just in line with the purpose of this study, so the results of the coupling coordination degree model are adopted for surveying and mapping rather than statistical methods. And the coupling coordination degree model itself is the most widely used model to measure the relationship between two systems.

3) Thanks for the very helpful comment. We have adjusted the structure of the revised manuscript. We have added a discussion section, in which we review the work done by the study, then list the strengths and weaknesses of the index, and finally look ahead to future work. The above discussion is added in the revised manuscript as Section 7, as highlighted by the yellow background.

specific comment

The most specific comments are the grammar, which should be checked by an English expert. The English is not my first language, but I still found lots errors in the grammar logic. For example, in the first two sentences, CI should be fully written down when it occurrs in the first time.

R: Thanks for the good suggestion. In the process of revision, we carefully re-check our writing. In order to further ensure the correctness of the grammar, we also hired relevant English proofreading companies to check and process the grammar by their English experts. In this regard, we asked the company and the experts to issue an English polishing certificate, which is attached for your reference.

Table 6: The first-level indicators of II. The indicator weight seems wrong? 12.78% should be 10.7?%?

R：In order to find out the problem of index weight, we recalculated and checked all the data in the study, and found that there was a problem in a value of the third-level index, which has been modified. In order to ensure the correctness of this result, we repeated the calculation three times. When the results of the three times were consistent, we conducted horizontal verification again to confirm the correctness of the submitted manuscript.

Figure 1: I believe the figure 1 should be re-draw following the instruction of the PLos ONE.

R：For Fig 1, we redrew it after reading the PLOS ONE instructions and some of the PLOS ONE published papers, and have replaced the Fig 1 in the revised manuscript.

Fig 1. Coupling Degree Sequence of CI and STI

Reviewer #2: This manuscript attempted to explore the correlation between Chinese cultural industry and scientific and technological innovation. Unlike previous studies that mainly took a qualitative approach, this study attempted a quantitative approach to this relationship. It is also encouraging that the authors presented various literature on the relationship between the cultural industry and science and technology innovation. Exploring the relationship between cultural industries and science and technology innovation in quantitative perspective seems interesting, but challenges involved in estimation must also be taken seriously.

1. Authors should be careful not to interpret correlations as causal relationships throughout the manuscript. For example, expressions implying a causal relationship, such as “the effect of a specific STI index on CI,” should be avoided and modified to indicate a correlation.

R：Thank you for your comments. The grey correlation model we adopted is indeed only used to describe the correlation between two systems rather than the causation relationship. We have modified this in the return draft, and the expression that may imply causation in the text has been modified to correlation.

2. The author used EVM, CCD, and GCD for empirical analysis.

However, this manuscript lacks an explanation of whether this analytical process is valid in the context of this analysis.

Authors are requested to provide detailed examples of similar analyzes performed using this methodology.

R：This is the first time to use CCD and GCD models to explore the micro and macro relationship between cultural industry and scientific and technological innovation. However, coupled coordination degree and grey correlation degree are also used to investigate the macro and micro relations between the other two systems. This method has been used by Chu et al.[70] to study the coordinated development of logistics industry and financial industry. More studies are used in environmental protection and sustainable development direction [71] or logistics industry development direction [72]. The above distance examples have been added to the chapter "Methodology" in the manuscript.

3.Explain how the authors controlled for the third confounder in the empirical estimate of the correlation.

R：We have tried our best to learn how to control confounding factors, because we have found no examples of confounding factors control in all known literatures using entropy method, coupling coordination degree model and grey relational degree model. But unfortunately, we are still confused about how to control confounding factors in this study. Two groups of research objects are needed to control confounding factors, and each index system in this study has only one group of objects. In the two index systems constructed in this study, indicators at all levels (such as level 1 and level 2, level 2 and level 3) belong to different levels, and the indicators at the first level are synthesized into the indicators at the upper level according to weight, which is a relationship between superiors and subordinates rather than an equal relationship. Therefore, there would be no imbalance in the distribution of prognostic factors between the treatment group and the control group, which would lead to the underestimation or overestimation of the true effect value, resulting in bias. In other words, there would be no confounding situation in this study, so we did not control confounding factors.

4.Sufficient evidence needs to be presented to justify the findings of the analysis.

R：The analysis results in this paper are based on empirical and actual data. In the revised manuscript, we expanded the content of the analysis and added relevant proofs to each analysis result, such as stating the reality or similar views of existing studies, so as to provide sufficient evidence to prove the correctness of the analysis results. The above discussion is added in the revised manuscript, as highlighted by the yellow background. 

5.In order to secure the reliability of the indicator, it is necessary to describe in more detail other cases where this indicator is used.

R: The index system constructed in this study is obtained according to the actual situation after reading a lot of previous literatures and referring to the indicators of related reports released by the government. For example, an indicator of the index system of cultural industry is based on reference to previous research experience , on the basis of comprehensive reference to Chinese Provincial and Municipal Cultural Industry Development Index 2012 、China Cultural Industry High-quality Development Index 2019 and other publicly released domestic authoritative cultural industry development index research, The evaluation index system of high quality development of cultural industry is constructed from four dimensions of industrial innovation, industrial resources, industrial opening and industrial sharing. Based on these studies, combined with the CI development index of Chinese provinces and cities in 2012 and the CI Quality Development Index of China in 2019, a first-level index of CI quality development evaluation index system was constructed from four aspects: industrial innovation (II), industrial resources (IR), industrial opening (IO) and industrial sharing (IS).

The Index system of science and technology Innovation is built by integrating published literatures. For example, the index system of Global Innovation Index 2018 published by the World Bank is slightly different from the index system built by China. But generally speaking, it is mainly constructed from the same four aspects of scientific and technological innovation environment, scientific and technological innovation input, scientific and technological innovation output, and scientific and technological innovation benefit. This study also uses this first-level classification in the construction of the index system. And a lot of reference to the index system built by predecessors, to set up a secondary index, the scientific and technological innovation evaluation system of this study.

In order to make readers more intuitive and convenient to understand other situations of using this index, a separate column is added after the index system to express other situations of using this index. The above discussion is added in the revised manuscript in Table 1, as highlighted by the yellow background. 

6. Between conclusions and recommendations The logical connection between conclusions and recommendations needs to be strengthened. To this end, it is necessary to supplement the recommendations based on the results of analyzing the relationship between CI and STI.

Considering the title and design of this study, it is necessary to make recommendations based on the relationship between the cultural industry and STI, not the cultural industry alone.

R: Thanks for this very helpful comment. The recommendations section has been rewritten to make recommendations strictly based on the conclusions. For example, the first suggestion is put forward according to the three endogenous conditions affecting the cultural industry; the second and third suggestions are put forward from the perspective of the strong correlation between scientific and technological innovation environment, scientific and technological innovation output and the development of cultural industry; the last suggestion is put forward according to the strong correlation between scientific and technological innovation and cultural industry. The logical connection between conclusions and recommendations has been strengthened, and suggestions have been made on the integration of scientific and technological innovation and cultural industry. 

The details are as follows:

(1) Increase spending on industrial resources and cultural programs and increase policy support. Industrial resources, policy support and cultural spending have played a decisive role in the development of the cultural industry. Therefore, we need to strengthen the guidance and support at the national level, combine the "visible hand" with the "invisible hand", give play to the leading role of macro-control on the basis of respecting the law of market economy, and support the development of the cultural industry with active industrial policies. On the one hand, the state can introduce relevant policies and measures to provide favorable treatment in terms of capital and taxes to cultural industry enterprises, increase capital investment, facilitate development, and provide development guarantees from the system, according to market demand and development trends. On the other hand, control and supervision should be strengthened in response to the endless cultural infringement incidents in recent years. 

(2) To create a good environment for STI and influence the development of the CI to better and faster development. (1) Encourage cultural and creative subjects (including cultural and creative enterprises and individuals) to carry out content and scene innovation. Fully excavating Chinese traditional culture and literary creation in line with the contemporary development trend, combine the traditional (historical) with The Times, so as to enrich the cultural content with novelty and The Times. We should increase the cultivation of high-level cultural talents, improve the "people-to-people bond" through international exchanges, introduce and absorb foreign excellent cultural forms and contents while going out, and attract outstanding cultural people to participate in cultural exchanges and innovation.

(3) We should attach importance to the influence of STIO on the development of CI, and promote the development of cultural industry by optimizing the carrier. Increase investment in cultural and technological infrastructure, including the construction of buildings, venues, studios and other aspects. The development of CI and STI requires the support of carriers. In the aforementioned research, S&T museums and related S&T activities have become key factors in promoting the coupled development of CI and STI. Therefore, it is necessary to increase investment in related infrastructure to provide material venues for the coupled development of CI and STI. Actively adopt new technologies, including 5G technology and big data technology, to better design, present and market cultural content. Invest in research and development, guide cultural and scientific personnel to cooperate and jointly develop, actively promote scene innovation and use STI to make the CI more attractive.

(4) Vigorously promote STI and promote the rapid development of the CI. The development of STI has a certain synergistic effect on the CI, so we need to seize this main "booster", give full play to the promoting role of STI in the development of CI, implement the strategy of "strong S&T" driving "strong culture", and vigorously develop emerging S&T. Seize the commanding heights of S&T to realize the leapfrog development of our CI, so as to overtake the curve and build a cultural power. For example, we should vigorously develop 5G technology, lay out the meta-universe, and realize the rapid development of the CI by creating new growth poles, developing new business ecosystems and business forms, and improving the transmission of cultural products. 

The above discussion is added in the revised manuscript as Section 8.2, as highlighted by the yellow background.

[Minor Points]

1. The 'Recommendations' section is described in the order of (1) (4) (5). It needs to be modified to (1) (2) (3)

R: Thanks for the remind, the serial number of the suggested part has been re-marked.

In order to meet the requirements of the PLOS ONE style, we have also made some modifications in format and presentation.

---

## [Decision Letter · Decision Letter 1]

27 Mar 2023

Evaluation of the Coordinated Development between Chinese Cultural Industry and Scientific & Technological Innovation

PONE-D-22-34472R1

Dear Dr. Yu,

We’re pleased to inform you that your manuscript has been judged scientifically suitable for publication and will be formally accepted for publication once it meets all outstanding technical requirements.

Kind regards,

Baogui Xin, Ph.D.

Academic Editor

PLOS ONE

Additional Editor Comments (optional):

Reviewers' comments:

Reviewer's Responses to Questions

**Comments to the Author**

1. If the authors have adequately addressed your comments raised in a previous round of review and you feel that this manuscript is now acceptable for publication, you may indicate that here to bypass the “Comments to the Author” section, enter your conflict of interest statement in the “Confidential to Editor” section, and submit your "Accept" recommendation.

Reviewer #1: All comments have been addressed

Reviewer #2: All comments have been addressed

2. Is the manuscript technically sound, and do the data support the conclusions?

Reviewer #1: Yes

Reviewer #2: Partly

3. Has the statistical analysis been performed appropriately and rigorously? 

Reviewer #1: Yes

Reviewer #2: N/A

4. Have the authors made all data underlying the findings in their manuscript fully available?

Reviewer #1: Yes

Reviewer #2: Yes

5. Is the manuscript presented in an intelligible fashion and written in standard English?

Reviewer #1: Yes

Reviewer #2: Yes

6. Review Comments to the Author

Reviewer #1: After reading the revised manuscript, I believe authors have addressed all comments from me. Thanks a lot. I suggest to accept this manuscript. However, some typo still be needed to be checked at the following stages.

Reviewer #2: It is encouraging that the authors have made significant improvements in writing and logical structure, but as a reviewer, I wish it had been based on a more robust statistical methodology. However, I believe that this study has value as an exploratory study.

7. PLOS authors have the option to publish the peer review history of their article (what does this mean?). If published, this will include your full peer review and any attached files.

Reviewer #1: No

Reviewer #2: No

---

## [Editor Report · Acceptance letter]

13 Apr 2023

PONE-D-22-34472R1 

Evaluation of the Coordinated Development between Chinese Cultural Industry and Scientific & Technological Innovation 

Dear Dr. Yu:

I'm pleased to inform you that your manuscript has been deemed suitable for publication in PLOS ONE. Congratulations! Your manuscript is now with our production department. 

Kind regards, 

on behalf of

Professor Baogui Xin 

Academic Editor

PLOS ONE